# 'Explaining RL Decisions with Trajectories':
# A Reproducibility Study

**Karim Abdel Sadek, Matteo Nulli, Joan Velja and Jort Vincenti**

University of Amsterdam *
**Reviewed on OpenReview:** https://openreview.net/forum?id=QdeBbK5CSh

## Abstract

This work investigates the reproducibility of the paper " Explaining RL decisions with trajectories " by Deshmukh et al. (2023). The original paper introduces a novel approach in explainable reinforcement learning based on the attribution decisions of an agent to specific clusters of trajectories encountered during training. We verify the main claims from the paper, which state that (i) training on less trajectories induces a lower initial state value, (ii) trajectories in a cluster present similar high-level patterns, (iii) distant trajectories influence the decision of an agent, and (iv) humans correctly identify the attributed trajectories to the decision of the agent. We recover the environments used by the authors based on the partial original code they provided for one of the environments (*Grid-World*), and implemented the remaining from scratch (*Seaquest* and *HalfCheetah*, *Breakout*, *Q\*Bert*). While we confirm that (i), (ii), and (iii) partially hold, we extend on the largely qualitative experiments from the authors by introducing a quantitative metric to further support (iii), and new experiments and visual results for (i). Moreover, we investigate the use of different clustering algorithms and encoder architectures to further support (ii). We could not support (iv), given the limited extent of the original experiments. We conclude that, while some of the claims can be supported, further investigations and experiments could be of interest. We recognize the novelty of the work from the authors and hope that our work paves the way for clearer and more transparent approaches.

## 1 Introduction

Reinforcement Learning (RL), formalized in the pioneering work of Sutton & Barto (2018), focuses on learning how to map situations to actions, in order to maximize a reward signal. The agent aims to discover which actions are the most rewarding by testing them. This addresses the problem of how agents should learn a policy that takes actions to maximize the cumulative reward through interaction with the environment. A recent pivotal focus in RL is the increasing attention on the explainability of these algorithms, a factor for their adoption in real-world applications. Precedent work in the field of XRL include Puiutta & Veith (2020), Korkmaz (2021) and Coppens et al. (2019). This reproducibility report focuses on the work of Deshmukh et al. (2023), which proposes an innovative approach to enhance the transparency of RL decision-making processes. Given the rising interest and applications of Offline RL (Levine et al. (2020),Kumar et al. (2020)), obtaining explainable decision is an important desideratum. Deshmukh et al. (2023) introduces a novel framework in the offline RL landscape. This new approach is based on attributing the decisions of an RL agent to specific trajectories encountered during its training phase. It counters traditional methods that predominantly rely on highlighting salient features of the state of the agent(Iyer et al. (2018)).

Despite proposing a novel approach, no subsequent work has built upon it. We believe this is primarily due to the absence of code available online and the limited number of environments on which this method was tested, which reduces its practical applications. Therefore, we intend to not only validate the original results

---

*{karim.abdel.sadek, matteo.nulli, joan.velja, jort.vincenti}@student.uva.nl

by Deshmukh et al. (2023), but also explore the robustness and generalizability of the proposed methodology across different environments and settings. Our reproducibility study is a step toward ensuring that advancements in this domain are both transparent and trustworthy, paving the way for a better understanding of RL systems.

## 2  Scope of Reproducibility

Explainability and interpretability have recently become of great interest for the adoption of AI systems in real-world applications. In particular, understanding and explaining the behavior and decisions of RL agents is a crucial task considering the plausible large-scale adoption of these systems. On top of the aforementioned ones in Section 1, other examples of Explainable Reinforcement Learning (XRL) studies include a high-level decision language approach by Puri et al. (2019), Pawlowski et al. (2020) and Madumal et al. (2020). The goal of this report is to analyze the reproducibility of the work by Deshmukh et al. (2023). Given the novelty of the work, it follows that there is no existing benchmark to compare the results claimed by the authors. Our contribution lies mostly in the implementation, verification, and interpretation per se of these results. We proceed by verifying the claims made by the authors, which we summarize and re-state as follows here below:

- ***Removing Trajectories induces a lower Initial State Value:*** Including all relevant trajectories in the training data will result in higher or equal initial state value estimates compared to training sets where key trajectories are omitted. This holds also for other metrics we may consider. The definitions can be found in Section 3.3.

- ***Cluster High-Level Behaviours:*** High-level behaviours are defined as patterns within a trajectory which lead to the same result and repeat over multiple trajectories. We aim to verify that different embedding clusters represent different meaningful high-level and interpretable behaviors.

- ***Distant Trajectories influence Decisions of the Agents:*** Decisions performed by RL agents can be influenced by trajectories distant from the state under consideration. In such scenarios looking only at the features in the action space may not provide a full understanding of the behaviour of an agent.

- ***Human Study:*** Humans may accurately identify the determinant trajectories that influenced the decision of an RL agent.

## 3  Methodology

The original paper code is not yet publicly available. However, we obtained part of the code from the authors: we were given the *Grid-World environment*, together with part of its related experiments. We followed their code and expanded upon it, in order to verify the claims. On the other hand, we wrote from scratch the implementation for *Seaquest, HalfChetaah*. Additionally, we added two environments to the analysis of Deshmukh et al. (2023), *Breakout* and *Q\*Bert*.

### 3.1  Environments

The investigations made in the paper regard three different types of Reinforcement Learning environments:

1. *Grid-World*, a grid-like environment in which the agent has a discrete state and action space. The game consists of an agent starting from a point in the grid and moving inside it. The goal is to reach a 'green' cell while avoiding entering a 'red' cell, while making the smallest number of steps possible. The default grid has a size of 7x7.

2. *Seaquest*, a video-game-like environment in which the agent has a discrete state and action space. The game consists in a submarine moving underwater. Here are more information on the Atari Seaquest environment.

3. *HalfCheetah*, a video-game-like environment where the agent has a continuous state and action space. The game consists of a 2-dimensional robot having a number of joints. The goal is to get the cat shaped robot to run, by turning its joints using rotational forces (torque). Here are more information on the *HalfCheetah* environment.

4. *Breakout*, one of the most played atari-produced video-games. The game consists of a paddle and hitting a ball into a brick wall at the top of the screen, where the goal is to destroy all the bricks. Here are more informations on *Breakout* environment.

5. *Q\*Bert* is a video-game like environment. The game consists of an agent which hops on each cube of a pyramid one at a time, and the goal is to change their color of into a specific one. Here are more informations on *Q\*Bert* environment.

## 3.2 Datasets

Datasets used in the analysis have the same composition between the five environments. Each dataset $D$ comprises of a set of $n_\tau$ trajectories. Each $\tau_j$ is a k-step trajectory and each trajectory step is a tuple $\tau_j = [\tau_{j,1}, \tau_{j,2}, ..., \tau_{j,k}]$ where $\tau_{j,i} = (o_i, a_i, r_i)$. Here $o_i$ is the observation in that step, $a_i$ is the action taken in that step and $r_i$ is the per-step reward. However, collecting data depends on the environment in which the experiments are made. Regarding *Grid-World*, agents are trained specifically to generate data trajectories. For *Seaquest* data is instead downloaded from d4rl-Atari Repository, for *Breakout* and *Q\*Bert* from Expert-offline RL Repository, whereas in the case of *HalfCheetah* from d4rl Repository of Fu et al. (2020).

## 3.3 Model Description

Across all three environments, trajectory attributions and explanations are made through the following 5 steps, also summarized in Figure 1:

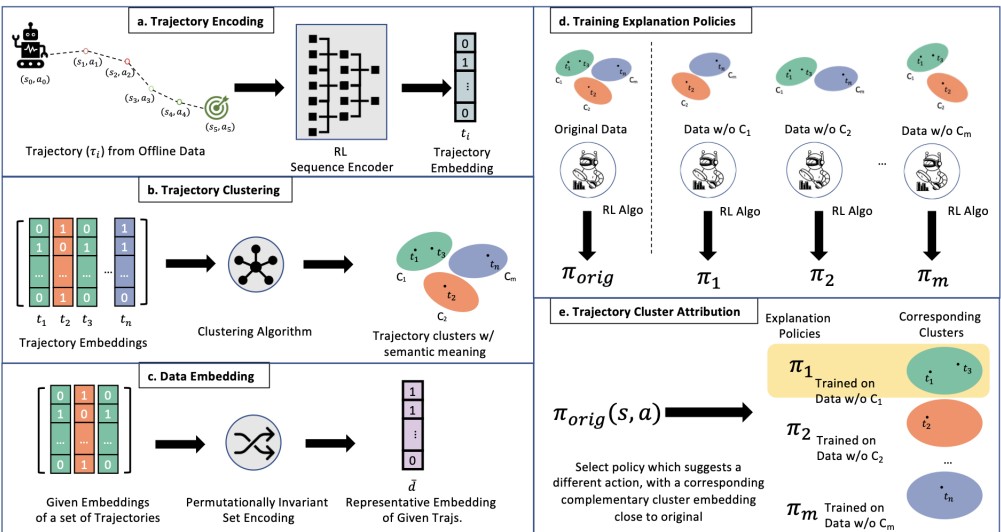

Figure 1: **Trajectory attribution process by Deshmukh et al. (2023)**

a. In *Grid-World* trajectories are generated by training different agents using Model-based offline RL through the *Dyna-Q Algorithm* (Appendix A.1). Trajectories are then encoded. In *Grid-World* the authors define a Seq2Seq LSTM based encoder-decoder architecture. After training, only the output of the encoder which corresponds to the *trajectory embedding* of Figure 1 is kept. On the

other hand, in all others ( *Seaquest*, *Breakout*, *Q\*Bert* and *HalfCheetah*) the trajectories encoders are pre-trained. For the former, the model is obtained following the instructions on pre-trained decision transformer. For the latter, the pre-trained model is downloaded from the GitHub repository pre-trained trajectory transformer from Janner et al. (2021). Both architectures are *GPTs*. Last but not least, these encodings are then embedded.

b. The *embeddings* are passed through the *XMeans* clustering algorithm introduced by Pelleg et al. (2000). The implementation used by the authors is the one from Novikov (2019). Using *XMeans* is an arbitrary choice and in Section 4.5 we will investigate other options.

c. The *cluster representations* are embedded obtaining the *representative embedding of given trajectories*.

d. The so-called complementary datasets are obtained. That is, for each cluster we create a different dataset where for each cluster $j$ we retain all the data but those trajectories belonging to cluster $j$ itself. We obtain then 10, 8, and 10 complementary datasets for the three environments respectively, and train for each complementary dataset new explanation policies and actions. In particular for *Seaquest*, *Breakout* and *Q\*Bert* we use *DiscreteSAC* Christodoulou (2019), whereas for *HalfCheetah* we employ *SAC* Haarnoja et al. (2018). They are state-of-the-art Reinforcement Learning algorithms merging Q-Learning with policy-optimization, used following the d4rl implementation by Seno & Imai (2022).

e. In the end, the decision made in a given state is attributed to a trajectory cluster.

Evaluation of new policies and actions is done through 5 metrics:

- *Initial State Value Estimate (ISV)* $\cdot$ $\mathbb{E}(V(s_0))$
  This metric is measuring the expected long-term returns in evaluating offline Reinforcement Learning training. The higher the ISV value, the better our trained policy.

- *Local Mean Absolute Action-Value Difference* $\cdot$ $\mathbb{E}[|\Delta Q_{\pi_{\mathrm{orig}}}(s)|]$
  Estimating how much the original policy differs from the new calculated one. High values are desirable.

- *Action Contrast Measure* $\cdot$ $\mathbb{E}[1(\pi_{\mathrm{orig}}(s) \neq \pi_j(s))]$
  Quantifying the disparity between recommended actions coming from the new explanation policies and original actions. Higher values are associated with better policies.

- *Wasserstein distance* $\cdot$ $W_{\mathrm{dist}}(\bar{d}, \bar{d}_j)$
  It measures the distance over a metric space between the original data embedding set and the complementary data embedding sets.

- *Cluster attribution frequency* $\cdot$ $\mathbb{P}(c_{\mathrm{final}} = c_j)$
  Computes the probability of a cluster being responsible one for an RL decision.

A low *Wasserstein distance* and high *Action Contrast Measure* values correspond with a higher frequency attribution.

### 3.4 Hyper-parameters

In order to reproduce the experiments of the paper we strictly used, when available, the same hyperparameters used by the authors. This was the case for *Grid-World*. Regarding *Seaquest*, *Breakout*, *Q\*Bert* and *HalfCheetah* we developed the code from scratch. Hence, we cannot be certain about the exact hyper-parameters used by the authors. In all instances, we retained the default settings provided by the libraries. If certain essential values were absent, we chose those that aligned with the settings used in *Grid-World*.

### 3.5 Experimental Set up and Code

Our experimental setup follows the approach of Deshmukh et al. (2023) in proving their claims. For claims *Clustering High Level Behaviour* and *Distant Trajectories influence Decisions of the Agent* we visually inspect the trajectories by plotting them, together with additional experiments. *Removing Trajectories induces a lower IVS* claim is carefully taken care of by inspecting the 5 different metrics previously introduced. *Human Study* claim is verified by replicating the analogous human study.

## 4 Results

In order to verify the previously stated claims, we proceed with an empirical evaluation, aiming to reproduce the results obtained by the authors. The result from Grid-Word presents some variance with respect to the ones reproduced in this article. Concerning the four other environments, they lack reproducibility due to a total absence of original code. Additionally, we further investigate each claim by conducting additional experiments. Most of these are done using the proposed number of trajectories by Deshmukh et al. (2023) (60 in *Grid-World* 7x7, 717 in *Seaquest*, 1000 in *HalfCheetah*), unless specified otherwise.

### 4.1 *Removing Trajectories induces a lower Initial State Value*

**Reproducibility:** In the *Grid-World* environment, the authors introduced different metrics to show that trajectories play an important role in obtaining high-quality policies. The values obtained by the authors are reported in Table 1 of the original paper. We present our findings in Table 1, providing evidence for the claim of the authors. The results are in general reproducible, with a small variation. The original policy, trained on all the trajectories, achieves the highest ISV among all. We report the reproduced results for the *Seaquest* environment in Table 2. We observe that the results are not similar to the ones in the original papers. This discrepancy could be attributed to several factors. The setup process, involving the installation of numerous packages and the use of outdated libraries, likely introduced minor computational variances. Additionally, changes in game versions, such as upgrading *Seaquest* from v4 to v5, affected the game-play dynamics, increasing the available action space. The choice of the difficulty level of the game also influenced the dataset, as easier versions had shorter trajectories due to quicker game terminations. Another motive for this is given by the limited amount of training we carried for our agent. In fact, given computational limitations, we are not able to train for a long horizon of time. Moreover, the settings of the experiments from the authors are unclear. There is no notion or explanation for what it means to train an agent until saturation, and no further details on the hyperparameters of the experiments (for additional details, see Appendix D). On the other hand, even given the differences and limitations explained above, the Claim *Removing Trajectories induces a lower Initial State Value* still holds. The policy trained on the whole dataset achieves a higher ISV than any of the other ones. At the same time, the other metrics are consistent in terms of conclusions we can draw. Summarizing, the difference in reproducibility is due to the limited details given by the authors, together with the limitation on our computational resources. Results for *HalfCheetah* are reported in Table 6.

We perform a further analysis to highlight the reproducibility of the results. To have a better comparison of our results with the one obtained by the authors, we report in Table 3 the average values for the metrics we are investigating. We report an average across the clusters. We found (minimal) differences in our results. The ISV for the original policy coincides to one of the original paper. We neglect the last column since all probabilities sum to one. This further supports the claim of the authors.

**Additional Experiments:** We further analyze if trajectories are important to obtain a good ISV. Specifically, we see if the clusters more often present in the attribution set hold a larger importance in influencing the ISV. Figure 2 shows an inverse correlation between the number of times a cluster was responsible for a change in the decision and the ISV of the policy trained without that cluster. This further validates claim *Removing Trajectories induces a lower Initial State Value*, providing additional insights. Extra details can be found in Appendix C.4.

| $\pi$ | $\mathbb{E}[V(s_0)]$ | $\mathbb{E}[\|\Delta Q_{\pi_{\text{orig}}}(s)\|]$ | $\mathbb{E}[1(\pi_{\text{orig}}(s) \neq \pi_j(s))]$ | $W_{\text{dist}}(\bar{d}, \bar{d_j})$ | $\mathbb{P}(c_{\text{final}} = c_j)$ |
|---|---|---|---|---|---|
| orig | **0.307** $\pm$ 2e-04 | - | - | - | - |
| 0 | 0.305 $\pm$ 9e-12 | 0.011 $\pm$ 0.001 | 0.041 $\pm$ 0.047 | 0.241 $\pm$ 0.227 | 0.000 $\pm$ 0.000 |
| 1 | 0.303 $\pm$ 2e-03 | 0.007 $\pm$ 0.011 | 0.041 $\pm$ 0.042 | 0.371 $\pm$ 0.348 | 0.025 $\pm$ 0.050 |
| 2 | 0.304 $\pm$ 0e+00 | 0.002 $\pm$ 0.001 | 0.122 $\pm$ 0.049 | **0.924** $\pm$ 0.150 | 0.0000 $\pm$ 0.000 |
| 3 | 0.304 $\pm$ 0e+00 | 0.022 $\pm$ 0.001 | 0.0000 $\pm$ 0.049 | 0.111 $\pm$ 0.094 | 0.0000 $\pm$ 0.000 |
| 4 | 0.305 $\pm$ 1e-11 | 0.041 $\pm$ 0.004 | 0.122 $\pm$ 0.049 | 0.142 $\pm$ 0.096 | 0.0000 $\pm$ 0.000 |
| 5 | 0.300 $\pm$ 3e-03 | 0.029 $\pm$ 0.005 | 0.041 $\pm$ 0.055 | 0.036 $\pm$ 0.017 | 0.175 $\pm$ 0.231 |
| 6 | 0.287 $\pm$ 5e-03 | **0.061** $\pm$ 0.015 | **0.163** $\pm$ 0.049 | 0.040 $\pm$ 0.051 | **0.500** $\pm$ 0.237 |
| 7 | 0.301 $\pm$ 7e-03 | 0.030 $\pm$ 0.016 | 0.020 $\pm$ 0.059 | 0.099 $\pm$ 0.178 | 0.150 $\pm$ 0.183 |
| 8 | 0.305 $\pm$ 6e-13 | 0.008 $\pm$ 0.010 | 0.021 $\pm$ 0.049 | 0.022 $\pm$ 0.026 | 0.100 $\pm$ 0.200 |
| 9 | 0.304 $\pm$ 1e-11 | 0.017 $\pm$ 0.011 | 0.143 $\pm$ 0.043 | 0.061 $\pm$ 0.095 | 0.05 $\pm$ 0.061 |

Table 1: **Quantitative Analysis and reproducibility study of Claim *Removing Trajectories induces a lower Initial State Value* for *Grid-World*.** Results show the mean $\pm$ standard deviation over 5 different seeds.

| $\pi$ | $\mathbb{E}[V(s_0)]$ | $\mathbb{E}[\|\Delta Q_{\pi_{\text{orig}}}(s)\|]$ | $\mathbb{E}[1(\pi_{\text{orig}}(s) \neq \pi_j(s))]$ | $W_{\text{dist}}(\bar{d}, \bar{d_j})$ | $\mathbb{P}(c_{\text{final}} = c_j)$ |
|---|---|---|---|---|---|
| orig | **3.569** $\pm$ 1e-04 | - | - | - | - |
| 0 | 3.537 $\pm$ 2e-02 | 0.217 $\pm$ 0.020 | 0.125 $\pm$ 0.050 | 0.008 $\pm$ 0.005 | 0.050 $\pm$ 0.000 |
| 1 | 2.679 $\pm$ 7e-9 | 0.869 $\pm$ 0.002 | **0.375** $\pm$ 0.280 | **0.933** $\pm$ 0.130 | 0.000 $\pm$ 0.000 |
| 2 | 2.858 $\pm$ 3e-01 | 0.880 $\pm$ 0.020 | 0.000 $\pm$ 0.000 | 0.071 $\pm$ 0.010 | 0.025 $\pm$ 0.050 |
| 3 | 3.061 $\pm$ 1e-02 | 0.625 $\pm$ 0.003 | 0.125 $\pm$ 0.163 | 0.190 $\pm$ 0.090 | 0.000 $\pm$ 0.000 |
| 4 | 2.310 $\pm$ 0e+00 | 1.700 $\pm$ 0.044 | 0.000 $\pm$ 0.000 | 0.005 $\pm$ 0.001 | 0.175 $\pm$ 0.231 |
| 5 | 2.701 $\pm$ 5e-06 | 0.650 $\pm$ 0.051 | 0.125 $\pm$ 0.183 | 0.002 $\pm$ 0.040 | **0.600** $\pm$ 0.256 |
| 6 | 2.010 $\pm$ 4e-02 | **1.697** $\pm$ 0.016 | 0.125 $\pm$ 0.163 | 0.010 $\pm$ 0.020 | 0.000 $\pm$ 0.000 |
| 7 | 2.664 $\pm$ 7e-03 | 1.018 $\pm$ 0.001 | 0.000 $\pm$ 0.000 | 0.001 $\pm$ 0.00 | 0.150 $\pm$ 0.100 |

Table 2: **Quantitative Analysis and reproducibility study of Claim *Removing Trajectories induces a lower Initial State Value* for *Seaquest*.** Results show the mean $\pm$ standard deviation over 5 different seeds.

Additionally, we report the ISV tables for the Q*Bert and Breakout games in Appendix section B.1, Tables 7 and 8. For Q*Bert, the highest ISV value is assigned to cluster 1, which invalidates the initial claim. On a brighter note, for Breakout, the claim holds. In the following three experiments, due to the lack of evidence for the other games, the experiments were discontinued for those two games. However, we report the shapes of the clusters in Appendix section B.3 for the sake of completeness.

### 4.2 *Cluster High-Level Behaviours*

**Reproducibility:** In *Grid-World*, this claim can be verified by either observing their shared high-level behavioural patterns or by using some *quantitative metric*. We deemed the latter to be a more appropriate approach. Thus, we proceeded to define this starting from inspecting trajectories belonging to that cluster and calculating the percentage of such manifesting a certain pattern. We show one trajectory for each of the three analyzed clusters in Figure 3. The following high-level behaviours are retrieved: *'Achieving Goal in Top right corner'*, *'Mid-grid journey to goal'* and *'Falling into lava'*. A practical explanation of this *self-defined metric* can be done analyzing the trajectory behavior of *'Falling into lava'*. This is spotted by looking for a -1 reward value in the last but one state, and then calculating the percentage of trajectories that have this wanted characteristic within each cluster. We repeat this for every cluster and pick those with a percentage value greater then 90%. The procedure is then iterated for the other two categories, by changing the characteristics to look for. In the *'Achieving Goal in Top right corner'* we look for a +1 reward in the last but one state and going towards position 6. Whereas in the *'Mid-grid Journey to goal'* category we look for trajectories starting in the middle of the grid and having a positive reward in the last but one

| $\pi$ | $\mathbb{E}[V(s_0)]$ | $\mathbb{E}[|\Delta Q_{\pi_{\text{orig}}}(s)|]$ | $\mathbb{E}[1(\pi_{\text{orig}}(s) \neq \pi_j(s))]$ | $W_{\text{dist}}(\bar{d}, \bar{d}_j)$ |
|---|---|---|---|---|
| Mean Clusters (Original Paper) | 0.3027 | 0.0231 | 0.0821 | 0.0301 |
| Mean Clusters(Reproduced) | 0.3029 | 0.0230 | 0.0714 | 0.1098 |
| $|\Delta|$ | 0.0002 | 0.0001 | 0.0107 | 0.0797 |

Table 3: **Additional Quantitative Analysis on claim *Removing Trajectories induces a lower Initial State Value* for *Grid-World.***

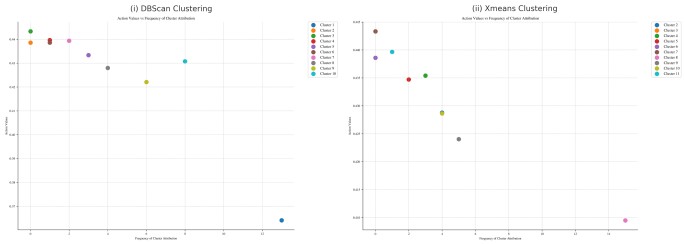

Figure 2: **Correlation between Action Value and the Cluster Attribution Frequency**. (i) The plot obtained using the DBSCAN algorithms shows a (weak) correlation of the action value with the attribution frequency of a cluster. We clearly observe that Cluster 1, which was the one attributed more often, is of crucial importance. (ii) The plot obtained using XMeans clearly shows the phenomena of Claim 2. There is a clear negative correlation between the two quantities, which highlights the importance of data trajectories. Again, the cluster attributed to most agent decisions, i.e. Cluster 7, constitutes a fundamental portion of the training data that leads to a high-value policy.

state. These align with the behaviours found by the authors. The claim is thus supported. Note that cluster labels vary from those highlighted by the authors.

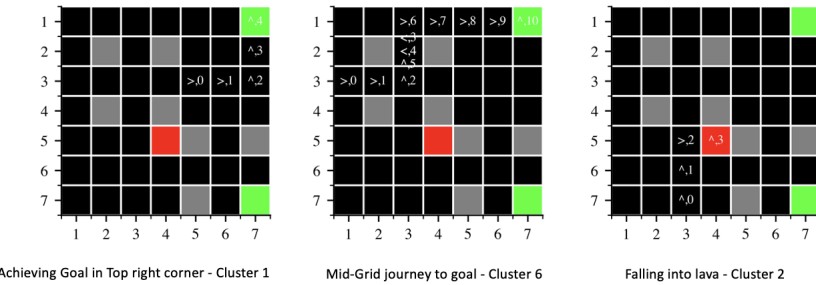

Figure 3: **Reproducing and verifying claim *Cluster High-Level Behaviours* in *Grid-World***. Cluster 1 showcases the presence of behaviour *'Achieving Goal in top right corner'*. Cluster 6 of *'Mid-grid journey to goal'* and cluster 2 of *'Falling into lava'*. Three High-Level Behaviours found match those highlighted by the authors.

In our *Seaquest* analysis, we tried to replicate the cluster findings from the original study in Figure 4. We noticed differences in the number of data points and their distribution. Converting 717 trajectories into around 24,000 sub-trajectories for the XMeans algorithm revealed more data points than shown in the original graph of the authors. This discrepancy could be due to two reasons: (i) the choice of game mode and data source might affect the length of observations, which was not detailed by the original authors, and (ii) the authors might have used a more complex method to aggregate data post-encoding than the simple averaging they described.

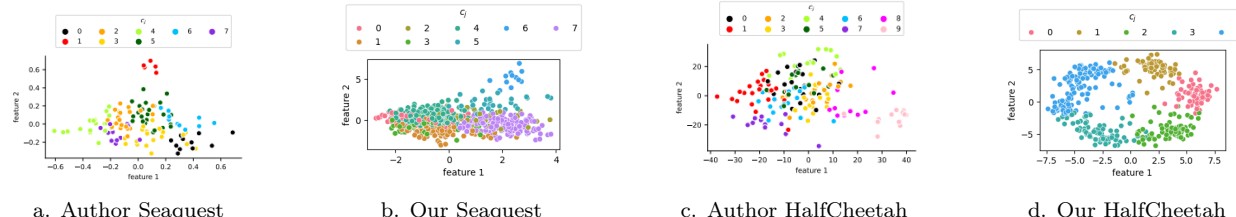

a. Author Seaquest        b. Our Seaquest        c. Author HalfCheetah        d. Our HalfCheetah

Figure 4: **Clustering differences in Seaquest and HalfCheetah**: This figure contrasts the clustering outcomes between our study and the original paper. Figure (a) and (c) illustrate the clusters of the authors for Seaquest and HalfCheetah, while figure (b) and (d) reflect our observations, revealing significant differences in distribution and amount of data points. These discrepancies may highlight the influence of game mode choices, dataset specifics, and data aggregation techniques on clustering outcomes.

Additionally, when trying to interpret the high-level meaning of those clusters, we obtained some discrepancies. Results in Figure 5 show a strong link between the *'Filling Oxygen'* behavior and cluster 7, while the other behaviors remained unclear, questioning the specific claims of the authors. However, this does not undermine the broader notion of 'meaningful clusters', but, given the scope of this paper, finding those interpretations was deemed excessively time-consuming and beyond our current objectives. More details can be found in Appendix B.2.

In the context of the *HalfCheetah* environment, assessing the significance of clusters proved challenging due to the high-dimensional and intricate nature of the observation space.

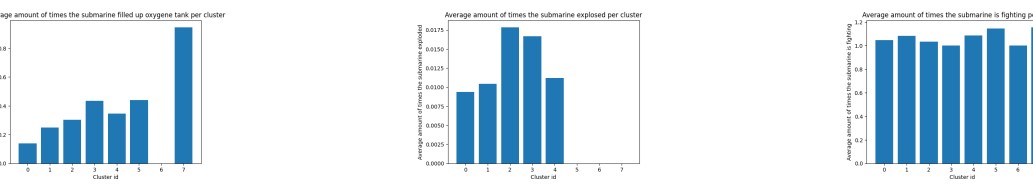

(a) Average time spent by each cluster filling their oxygen tanks

(b) Average number of submarine destructions per cluster

(c) Average surface combat encounters per cluster

Figure 5: **This figure evaluates high-level player behaviours in the clusters of the Seaquest game.** Sub-figure (a) shows the average time spent on filling oxygen, sub-figure (b) details the average submarine explosion, and sub-figure (c) counts surface combat encounters. These insights collectively enhance our understanding of the meaning within each cluster.

**Additional Experiments:** The three initial patterns explained in 4.2 are found both when using *XMeans* and *DBSCAN* and also when training with a higher number of trajectories (250). In this section, we investigate whether other meaningful high-level behaviours exist. We successfully identified an additional pattern. That is, each trajectory belonging to the same cluster has the same length. However, this last behaviour emerges only when using 60 trajectories with both *XMeans* and *DBSCAN*. It is not found when the number of trajectories used increases to 250. This phenomenon may arise due to the increased granularity of each cluster, a scale varying with the number of trajectories. Namely, by keeping the number of clusters fixed, less granularity is obtained. Thus each cluster can obtain worse cluster representations by grouping a larger number of trajectories together. However, this does not negate the claim of the authors because the original behaviours are also spotted with a higher number of trajectories.

### 4.3 *Distant Trajectories influence Decisions of the Agents*

**Reproducibility:** We start by analyzing claim *Distant Trajectories influence Decisions of the Agents* in the discrete *Grid-World* environment. The authors perform a qualitative analysis to support their claim. We try to reproduce the results and the plots given the code provided by the authors. The hyperparameters are set equal to the default values. The results of Figure 2 in the paper by Deshmukh et al. (2023) are reproducible

using the code given by the authors. Note that it may take more than one attempt to reproduce these results. This is due to the possible variation of clusters from each iteration of the code. Nevertheless, we found that these results were easily reproducible with little effort. Results are shown in Figure 6. The trajectories (i),(ii), (iii) are equivalent to the ones indicated in the paper by the authors. We plot an additional trajectory which is part of the attributed cluster. It is important to stress that also (iv) is distant from the state $(1, 1)$ we are considering. This investigation confirms the claim of the paper. However, given the highly qualitative justification provided by the authors, we seek a more structured and quantitative way of analyzing this claim. We defer these experiments to Section 4.3.

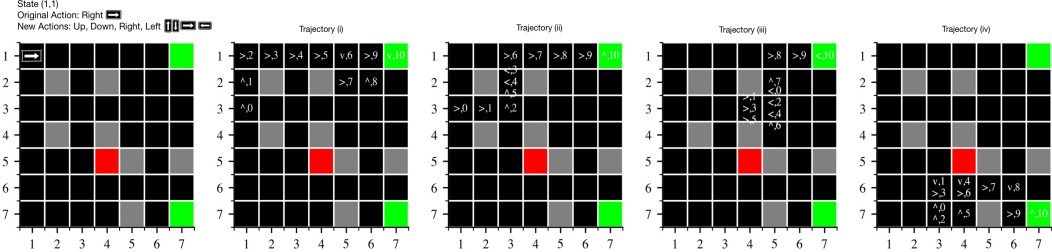

Figure 6: **Plot of grid and trajectories reproducing results and verifying Claim _DTAG_**. The original optimal action is 'right' in the state (1,1). When removing the trajectories belonging to the attributed clusters, all decisions are equally optimal, i.e. 'right', 'left',' up', or 'down'. This decision is attributed to 8 different trajectories, of which 4 plotted here above.

**Additional Experiments:** Reproducibility section 4.3 highlighted how trajectories far from our state of interest can influence the decision of our agent. However, it is not explicitly clear to what extent this is true. We strive to perform a more rigorous analysis, considering each state with attributed trajectories responsible for a decision change. For each of these attribution sets, we compute the average distance from the state to its trajectories. A rigorous definition of how we calculate the average and the distances can be found in Appendix C.3, together with a detailed pseudo-code in 1. In our experiments, two different cluster algorithms are employed. For _DBSCAN_ we set $\epsilon = 2.04$. Ten final clusters are obtained. No seed is needed given the deterministic nature of _DBSCAN_. The other cluster method is _XMeans_. The seed is set to 0 and 99 respectively for the initialization of the centers and for the _XMeans_ algorithm. We perform our experiments on the Grid-World Four Room Environment introduced by Sutton et al. (1999). Its size is 11x11. Given the larger grid and the scope of our experiment, we generate a higher number of trajectories. Namely, we produce 250 trajectories that end in a positive terminal and 50 trajectories that achieve a negative reward.

**Results**: Figure 7 shows the Bar-plot of these trajectories. We can observe that the average lengths are mostly larger than 6 in both settings. Interestingly, we note that using the _XMeans_ algorithm we have no state which is explained only by trajectories passing through it. From the results of the plot, we conclude that indeed distant trajectories are important to explain the decision of an RL agent, further confirming claim _Distant Trajectories influence Decisions of the Agents_.

### 4.4  _Human Study_

In order to verify this claim we reproduce the experiments as well as the study setup of the authors. Deshmukh et al. (2023) study is conducted on 10 people, which may not itself be sufficient to support the claim. In trying to improve this, we doubled the interviewees to 20 people, each of whom first received an explanation of how _Grid-World_ navigation works. Following this explanation, they all gained a full understanding of the navigation process. 40% are university graduates in mathematics and computer science. 45% are student graduates in engineering. The remaining 5% come from different study backgrounds. We begin by showing two Attributed Trajectories (_Attr traj 1_ and _Attr traj 2_), one _Random_ Trajectory, and one

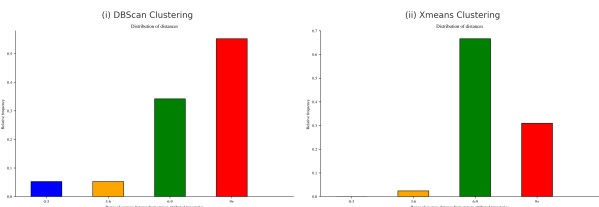

Figure 7: **Distances between states and their attributed trajectories**. On the x-axis, we have 4 bins, respectively in the range 0-3, 3-6, 6-9, and 9+. (i) shows the results using the *DBSCAN* algorithm. We see that we have many trajectories with a length bigger than 9. The states that fell in the blue bin were explained only by 1 or 2 trajectories. This is due to how *DBSCAN* constructs the clusters, which favors a big variance in size between them. We had then a bigger chance of having clusters with few trajectories that were all passing through an attributed state. (ii) shows the results using the *XMeans* algorithm. We note again that the majority of the states had a large average distance to their attributed trajectories.

Trajectory belonging to an *Alternate* cluster for each state. We investigate two questions. (i) Which single trajectory do you believe best explains the action suggested? (ii) Can you point out all the trajectories you believe explain the action suggested?

Averaging between both states, the results on Question (i) highlight how almost $\sim 72.5\%$ of the interviewees correctly identify one of the two attributed trajectories. For Question (ii), Figure 8 shows that in $\sim 63\%$ of the cases, humans can correctly identify all the attributed trajectories. The results obtained are similar to the ones of the authors. However, given the very limited sample size of our experiments, we do not have enough evidence to support the claim.

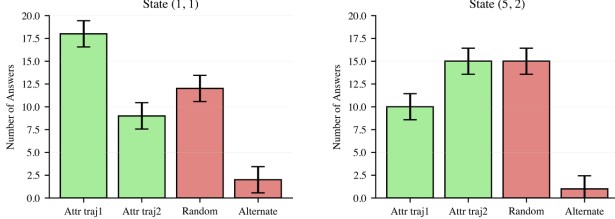

Figure 8: **Human Study**. The plot represents human answers to the two questions introduced. *Attr traj 1* and *Attr traj 2* are trajectories belonging to the cluster attributed for the decision of the agent in the specific state. *Random* stands for a randomly selected trajectory from the whole set. Whereas *Alternate* is a randomly selected trajectory from the whole set without those belonging to the attributed cluster. The results presented are for states (1,1) and (5,2). In both states we notice a decent level of human understanding. This suggests a meaningful understanding of which trajectories influence the agent's decision-making process.

### 4.5 Results beyond original paper

In this section we go beyond the author claims and try to experiment with the authors design choices like the employed encoder method as well as cluster algorithm.

**Improving clustering algorithm**: *XMeans* has proven to be useful in determining almost accurately the correct clustering trajectories, we propose a different approach by using *DBSCAN* algorithm. Introduced in Ester et al. (1996), *DBSCAN* is a non-parametric density-based clustering method that groups together sets of points packed together. The density-based characteristic of *DBSCAN*, differently to *X-Means*, is

not relying on the assumption that clusters are spherical-shaped. It is rather able to find clusters with any kind of variance between the points. This new algorithm could lead to new clusters, and possibly different insights from which we may be able to extract new information. **Results**: Figure 9 shows a *better visual clusters representation* computed by *DBSCAN*. This outcome is particularly visible in the reduced amount of overlaps which are instead present in *XMeans*. Additional metric results are available in Appendix C.

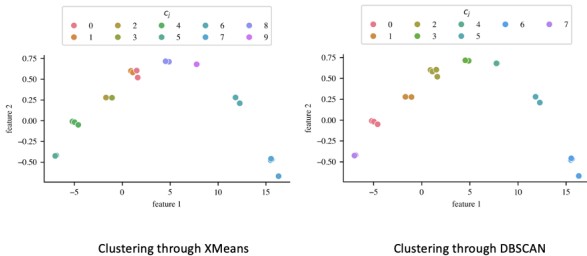

Clustering through XMeans          Clustering through DBSCAN

Figure 9: **Clustering Methods: *XMeans* vs *DBSCAN***. Through *DBSCAN* we obtain a lower number of clusters which eliminates overlaps between *XMeans* clusters 2, 3 and 0, 1. This *better visual representation* is mainly due to a difference in the algorithmic process.

**Different encoder techniques**: In *Grid-World* Environment the LSTM-based Seq2Seq encoding used by the authors has proven to be efficient. However, in this section we set out to experiment with different encoding techniques. Our hope is that they could provide a better hidden representation for trajectories. We employed two kinds of pre-trained encoders: Trajectory Transformer originally proposed in Janner et al. (2021). The model takes as input data in the form *(state, action, reward)* matching perfectly with the one provided by the authors. BERT base model Devlin et al. (2019). Pre-trained using Masked-Language Modelling, its capability of being adapted to many scenarios made it a good candidate to replace the LSTM in the paper. **Results:** Experiments are performed over 250 trajectories. We defer the table of results to 10, as we obtain no notable increase in performance across all metrics. Additionally, an inspection of high-level behaviors of clusters, as in section 4.2, highlights similar results.

## 5 Discussion

Across this work, we performed several experiments aimed at reproducing key findings of Deshmukh et al. (2023). The outcomes of this reproducibility study partially confirm their claims.

|  | Grid-World | Seaquest | Half-Cheetah | Breakout | Q*Bert |
|---|---|---|---|---|---|
| Removing trajectories | ✓ | ✓ | ✗ | ✓ | ✗ |
| Cluster behaviours | ✓ | ✗ | ✗ | ? | ? |
| Distant trajectories | ✓ | ? | ? | ? | ? |
| Human study | ? | ? | ? | ? | ? |

Table 4: **Summary**: Reproduced Results per Game for Each Claim. A ✓ represents validated results, a ✗ denotes an invalidated statement for the specific game, and ? indicates that we cannot confirm or deny the claim of the authors for this specific game. This may arise from time constraints or because the claim itself lacks sufficient precision, making it impossible to definitively confirm or refute even with additional experimentation.

**Reproducibility:** As reported in table 4, we can sustain claim *Removing Trajectories induces a lower Initial State Value* for *Grid-World*, *Seaquest* and *Breakout*. Our results for *HalfCheetah* and *Q*Bert* do not sustain this claim. The claim *Cluster High-Level Behaviours* is accepted only in *Grid-World*. In *Seaquest* we identified only one out of three high-level behaviors. This is not enough to support the claim. We were not able to obtain any high-level behaviour for *HalfCheetah*, *Breakout* and *Q*Bert*. On claim *Distant Trajectories influence Decisions of the Agents*, we obtained results consistent with the original ones, supporting the claim

for *Grid-World*. Our additional experiments further show that distant trajectories have a significant impact on the decision of the agent. The *Human Study* was carried out only in *Grid-World*. While we obtained similar results, the claim was superficially investigated by the authors. Their original experiments were not sufficient to support *Human Study*. Our survey, although more extensive, was limited due to time and resources constraints. As a consequence, we can not confirm *Human Study*.

**Code:** Across environments, reproducibility varies significantly due to code availability. The *Grid-World* code provided by the authors lacked a proper seeding mechanism. Despite this, similar reproducibility was not highly jeopardized. All of our experiments 4 are completely reproducible. First, in Seaquest and HalfCheetah the code was unavailable. Relevant implementation details, hyperparameters, and training techniques were not mentioned in the paper. We coded everything from scratch. This includes the whole trajectory attribution process, environment requirements, library dependencies, training and evaluation loops, and many more. In spite of these difficulties, we were able to perform the original experiments, supporting most of the claims of the authors when obtaining relevant evidence. We provide our complete code implementation. Additional training and implementation details can be found in Appendix D. The complete code implementation is available in our GitHub Repository.

**What was easy:** The authors shared the code for the *Grid-World* environment directly with us, although it is not publicly available online. This facilitated our ability to try to reproduce some of the results of the paper for this environment. In addition, the code provided was relatively easy to understand and adapt in order to expand our reproducibility study further. This allowed us to extensively produce additional experiments.

**What was difficult:** As mentioned throughout previous sections of the paper, the implementations for the Seaquest and HalfCheetah environments were not provided to us, thus leading to our implementation from scratch. Although very relevant to the results, the paper only briefly mentions *Additional Training Details* in the appendix, lacking any other explanation about the Python environment being used, any data pre-processing stage, or tweaks required for compatibility. With regards to the environment issues, it is worth noting that, being the field very novel and active, some of the libraries implemented do not get developed any longer. This may be the case of 'mujoco-py', the library on which HalfChetaah relies on: because of this, we had to backtrack compatibility between dependencies of libraries that are in constant development and some whose support has been interrupted for years. On top of this, we also encountered compatibility issues between different Operative Systems: MacBooks produced post-2020 abandoned Intel-based processors to adopt natively built Apple Silicon processors. This required the community to rebuild packages to support a new architecture (arm64 vs x86/x64), in order for these libraries to be able to run on these devices. Co-occurrently, support for Mujoco-py has been halted, thus requiring a somewhat convoluted installation procedure, consisting of manual pointers definition, third-party compilers requirement and many other details we decided not to bother the reader with. Needless to say, the lack of code in this delicate instance, required extensive trial and error experimentation with dependencies and installation procedures, thus slowing down the overall process for housekeeping operations rather than actual development. We believe the lack of transparency with regards to the aforementioned environments led to such striking differences in absolute value results between our reproduced results and the original ones: again, details on the training procedure and model implementation should have been made public, given the rather complex nature of the task at hand, along with hyperparameters setup, which we go over in further detail in Appendix D.

**Communication with the authors:** Communication with the authors was carried out by the course coordinators themselves at the beginning of the course. We received the aforementioned subset of the original implementation halfway through the course, thus leading to the issues discussed in the previous section. No further communication with the authors has been conducted by us.

**Key takeaways and original paper limitations:** In our investigation we find that the proposed methodology is of questionable effect, as it is not yet generalizable to many environments. Another key limitation is allowing only one cluster per attribution. We believe that allowing the method to consider trajectories from more then one cluster could lead to more a comprehensive analysis. Nonetheless we believe that this work Deshmukh et al. (2023) and its proposed attribution process have the possibility of unlocking new streams of research into a better understanding of Reinforcement Learning systems.

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

# A Methodology

## A.1 Model description

**Dyna-Q Algorithm**

Introduced in Sutton (1990), *Dyna-Q* is a Model based Reinforcement Learning Algorithm. Conceptually it is an algorithm that illustrates how real and simulated experience can be combined in building a policy. Dyna-Q algorithm (10) introduced in Section 3.3 starts by initializing a so-called Q table. A table made of all possible states vs all possible actions. The model also contains a *state*, *action*, *next state*, and *reward* tuples. This way the model can be both improved and queried to get the next state in the planning part. The process begins by observing state $S$ (a), and then selecting the next action $A$, in a greedy manner (b). After taking the action $A$, we observe a reward $R$ and a state $S'$. These two values are then used in the formula in (d) to update the Q table cell corresponding with state s and action a. After these classic Q-Learning steps we perform a loop (f) which consists of the added Dyna-Q part. First, we randomly select a state $S$ and an action $A$ and then we deduce a new state $S'$ and a new reward $R$ which will then be used to update the Q-table as before.

> **Tabular Dyna-Q**
>
> Initialize $Q(s, a)$ and $Model(s, a)$ for all $s \in \mathcal{S}$ and $a \in \mathcal{A}(s)$
> Loop forever:
>     (a) $S \leftarrow$ current (nonterminal) state
>     (b) $A \leftarrow \varepsilon$-greedy$(S, Q)$
>     (c) Take action $A$; observe resultant reward, $R$, and state, $S'$
>     (d) $Q(S, A) \leftarrow Q(S, A) + \alpha \left[ R + \gamma \max_a Q(S', a) - Q(S, A) \right]$
>     (e) $Model(S, A) \leftarrow R, S'$ (assuming deterministic environment)
>     (f) Loop repeat $n$ times:
>         $S \leftarrow$ random previously observed state
>         $A \leftarrow$ random action previously taken in $S$
>         $R, S' \leftarrow Model(S, A)$
>         $Q(S, A) \leftarrow Q(S, A) + \alpha \left[ R + \gamma \max_a Q(S', a) - Q(S, A) \right]$

Figure 10: **Dyna-Q algorithm**

## A.2 Computational Requirements & Environmental Impacts

The experiments were performed using a MacBook with Apple M2 Pro silicon chip with 10 CPU cores (1), MacBook with Apple M1 silicon chip with 8 CPU cores (2), and a Microsoft Windows 11 Pro with Intel(R) Core(TM) i7-10710U with 6 CPU cores (3). Most of the experiments in *Grid-World* run easily and in seconds on our local machines. On the other hand the running time for *Seaquest* and *HalfCheetah* with machine (1), can vary between 50 minutes (with 10 steps per epoch) and 8 hours (with 100 steps per epoch). We employed pre-trained models both for *Seaquest* and *HalfCheetah* as explained in Sections 3.3 and 3.2. The same is done for some additional experiments on *Grid-World* (Section C.2). Discussion on the environmental impact of these models has been addressed in previous literature (Rillig et al. (2023)). In addressing our own ecological footprint, we used the Code Carbon Tool (on machine (1)) to estimate our total energy consumption in obtaining cluster attributions. In *Grid-World* the estimated consumption is approximately 0.000170 kWh of electricity. Whereas in *Seaquest* we use 0.005133 kWh of electricity. Similarly, in *HalfCheetah* we consume 0.004783 kWh. We then use the $CO_2e$ equation to obtain the corresponding $CO_2$ emissions. The formula

$$CO_2e = CI \cdot PUE \cdot P \cdot t$$

is comprised of $CI$ Carbon Intensity (fixed value of 0.954), $PUE$ Power Usage Effectiveness (also fixes to 1.58), $P$ Power required (estimated through Code Carbon Tool) and the training time $t$ in hours. Our final emissions are available in Table 5.

| $\pi$ | Grid-World | Seaquest | HalfCheetah |
|---|---|---|---|
| $CO_2e$ lbs (10 steps per epoch) | 0.0000021 | 0.0064470 | 0.0060070 |
| $CO_2e$ lbs (100 steps per epoch) | - | 0.0618965 | 0.0576760 |

Table 5: **Emission levels in training our models from top to bottom**. The results highlight our CO2 equivalent levels in training the models. *Seaquest* and *HalfCheetah* were both trained using a two different number of iterations per epoch. However, *Grid-World* was not trained with a different configuration as a good performance was attained when using the standard hyper-parameters employed by the authors.

## B Results reproducing original paper and verification of the claims

### B.1 Removing trajectories induces a lower ISV

Despite the challenges in replicating the exact clustering outcomes for *Half Cheetah* as highlighted earlier, 6 reveals some interesting patterns. The table provides a quantitative analysis that, despite reproducibility issues, still shows consistent trends across different metrics.

| $\pi$ | $\mathbb{E}[V(s_0)]$ | $\mathbb{E}[\lvert\Delta Q_{\pi_{\mathrm{orig}}}(s)\rvert]$ | $\mathbb{E}[1(\pi_{\mathrm{orig}}(s) \neq \pi_j(s))]$ | $W_{\mathrm{dist}}(d, d_j)$ | $\mathbb{P}(c_{\mathrm{final}} = c_j)$ |
|---|---|---|---|---|---|
| orig | 3.3615 | - | - | - | - |
| 0 | 3.4558 | 0.0942 | 0.0038 | **1.0000** | 0.1000 |
| 1 | 3.4691 | 0.1076 | 0.0028 | 0.3047 | 0.3000 |
| 2 | 3.2958 | 0.0656 | 0.0035 | 0.8730 | 0.0000 |
| 3 | 3.3621 | 0.0006 | 0.0017 | 0.8483 | 0.0000 |
| 4 | 3.3624 | 0.0009 | 0.0022 | 0.2986 | **0.6000** |
| 5 | **3.5280** | **0.1665** | 0.0041 | 0.5340 | 0.0000 |
| 6 | 3.3444 | 0.0170 | 0.0016 | 0.5245 | 0.0000 |
| 7 | 3.3206 | 0.0408 | **0.0052** | 0.6162 | 0.0000 |
| 8 | 3.3745 | 0.0602 | 0.0028 | 0.6039 | 0.0000 |
| 9 | 3.3826 | 0.0337 | 0.0013 | 0.7855 | 0.0000 |

Table 6: **Quantitative Analysis and reproducibility study of Claim *Removing Trajectories induces a lower Initial State Value* for *HalfCheetah***

| $\pi$ | $\mathbb{E}[V(s_0)]$ | $\mathbb{E}[\lvert\Delta Q_{\pi_{\mathrm{orig}}}(s)\rvert]$ | $\mathbb{E}[1(\pi_{\mathrm{orig}}(s) \neq \pi_j(s))]$ | $W_{\mathrm{dist}}(d, d_j)$ | $\mathbb{P}(c_{\mathrm{final}} = c_j)$ |
|---|---|---|---|---|---|
| orig | 1.075 | - | - | - | - |
| 0 | 0.962 | **0.116** | **1.000** | 0.333 | 0.000 |
| 1 | **1.118** | 0.045 | 0.000 | 0.501 | **0.933** |
| 2 | 1.104 | 0.031 | 0.000 | **1.000** | 0000 |
| 3 | 1.006 | 0.0945 | 0.933 | 0.158 | 0.067 |
| 4 | 1.059 | 0.029 | 0.986 | 0.167 | 0.000 |

Table 7: **Quantitative Analysis and reproducibility study of Claim *Removing Trajectories induces a lower Initial State Value* for *Q\*bert***.

| $\pi$ | $\mathbb{E}[V(s_0)]$ | $\mathbb{E}[\lvert\Delta Q_{\pi_{\mathrm{orig}}}(s)\rvert]$ | $\mathbb{E}[1(\pi_{\mathrm{orig}}(s) \neq \pi_j(s))]$ | $W_{\mathrm{dist}}(d, d_j)$ | $\mathbb{P}(c_{\mathrm{final}} = c_j)$ |
|---|---|---|---|---|---|
| orig | **0.812** | - | - | - | - |
| 0 | 0.797 | 0.018 | 0.512 | 0.579 | 0.000 |
| 1 | 0.811 | 0.001 | 0.000 | **1.000** | 0.000 |
| 2 | 0.798 | 0.013 | 0.000 | 0.954 | 0.000 |
| 3 | 0.764 | **0.048** | **1.000** | 0.159 | 0.000 |
| 4 | 0.791 | 0.021 | 0.000 | 0.254 | **1.000** |

Table 8: **Quantitative Analysis and reproducibility study of Claim *Removing Trajectories induces a lower Initial State Value* for *Breakout***.

### B.2 Meaningful Clusters

In order to obtain Figure 5, we specifically analyzed the oxygen tank indicator at the bottom of the screen, recognizable by its unique color. An increase in the bar is interpreted as the submarine refilling its oxygen tank, while a empty tank resulted in a submarine explosion. This might occur either from running out of oxygen or sustaining damage from enemies. For the *'Fighting with head out'* behavior, we monitored the position of the submarine within the top 30 pixels of the screen. We would consider it as engaging in surface combat if it remained in this area for more than 10 out of the 30 frames in a sub-trajectory.

The analysis highlights distinct behaviors: Cluster 7 is linked to *'Filling Oxygen'*, and Clusters 2 and 3 to *'Submarine Burst'*, with no clear trend for *'Fighting with Head Out'*. 11 shows overlapping behaviors across clusters, complicating the attribution of specific actions. Consequently, the most representative cluster appears to be the one of refueling oxygen: two frames depict the submarine at the surface (directly implying oxygen refueling), while the remaining three suggest imminent game resets, indirectly associated with oxygen refill.

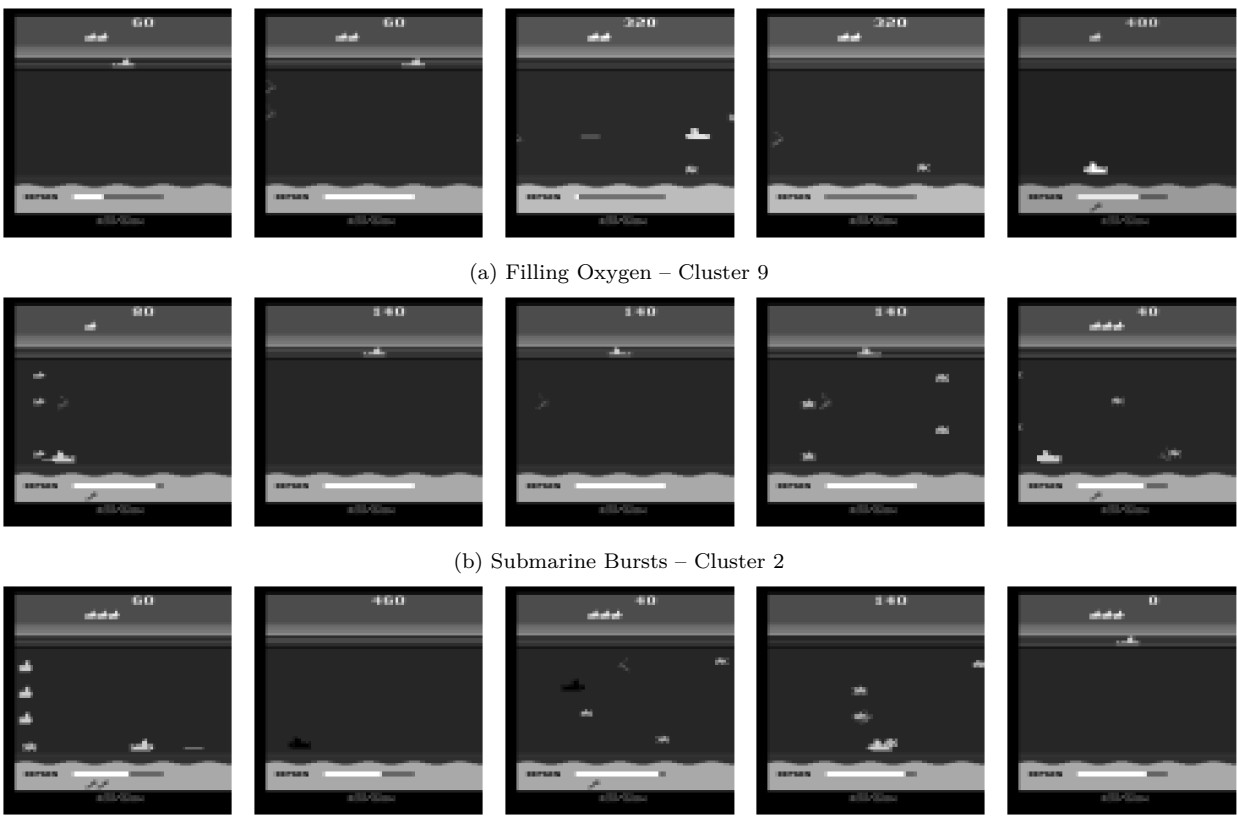

(a) Filling Oxygen – Cluster 9

(b) Submarine Bursts – Cluster 2

(c) Fighting with Head Out – Cluster 5

Figure 11: **Reproducing Figure 8 of the authors paper**. High-level Behaviours found in clusters for *Seaquest* formed using trajectory embeddings produced using decision transformer. The figure shows 3 example high-level behaviours along with the action description and id of the cluster representing such behaviour

### B.3 Clusters for Q*Bert and Breakout

Displayed in Figure 12 are the clusters for Q-Bert and Breakout. The choice of five clusters was made for both games, as they independently feature significantly more rewards within the trajectories as well as fewer action states (only 6 and 4, respectively), compared to their Seaquest counterpart. Consequently, we opted

to reduce the length of the sub-trajectories to 15 and to decrease the number of clusters to account for these differences.

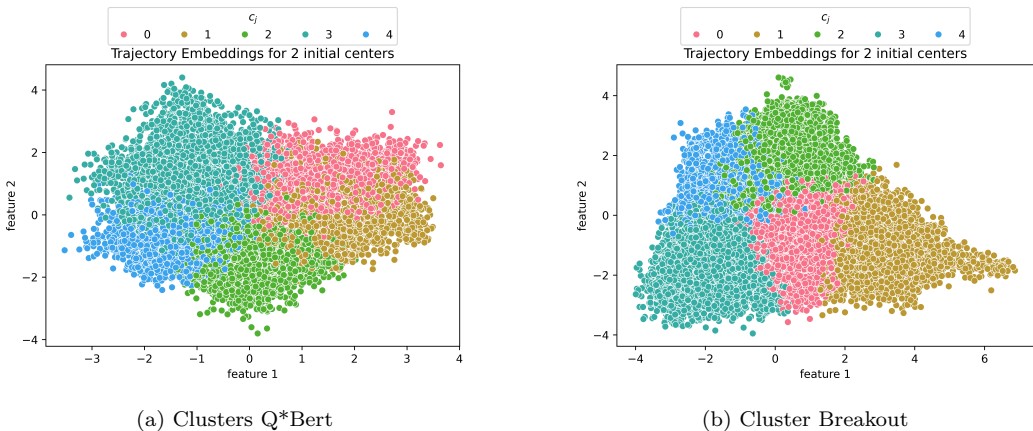

(a) Clusters Q*Bert

(b) Cluster Breakout

Figure 12: **Clusters** for the new games Q*Bert and Breakout

## C    Results beyond original paper

### C.1    Improving clustering algorithm

Table 9 is mirroring the results produced in Table 1. In this section however we produce metrics results using *DBSCAN* clustering algorithm, instead *XMeans*. As introduced before the values are similar to those attained in the original table.

| $\pi$ | $\mathbb{E}[V(s_0)]$ | $\mathbb{E}[\|\Delta Q_{\pi_{\text{orig}}}(s)\|]$ | $\mathbb{E}[1(\pi_{\text{orig}}(s) \neq \pi_j(s))]$ | $W_{\text{dist}}(\bar{d}, \bar{d}_j)$ | $\mathbb{P}(c_{\text{final}} = c_j)$ |
|------|------|------|------|------|------|
| orig | **0.3059** | - | - | - | - |
| 0 | 0.3056 | 0.0018 | 0.0613 | **1.0000** | 0.0000 |
| 1 | 0.2980 | 0.0326 | **0.1429** | 0.0009 | 0.1250 |
| 2 | 0.3045 | **0.0406** | 0.1225 | 0.0020 | 0.0000 |
| 3 | 0.3058 | 0.0281 | 0.0000 | 0.00007 | 0.0000 |
| 4 | 0.3045 | 0.0026 | 0.1021 | 0.0003 | **0.5000** |
| 5 | 0.3045 | 0.0288 | 0.1225 | 0.0005 | 0.0000 |
| 6 | 0.2859 | 0.069 | 0.0817 | 0.0018 | 0.3750 |
| 7 | 0.3058 | 0.0204 | 0.0205 | 0.0457 | 0.0000 |

Table 9: **Quantitative Analysis of *DBSCAN* algorithm**

### C.2    Implementing different encoder techniques

| $\pi$ | $\mathbb{E}[V(s_0)]$ | $\mathbb{E}[\|\Delta Q_{\pi_{\text{orig}}}(s)\|]$ | $\mathbb{E}[1(\pi_{\text{orig}}(s) \neq \pi_j(s))]$ | $W_{\text{dist}}(\bar{d}, \bar{d}_j)$ |
|------|------|------|------|------|
| Mean Clusters (Original Paper) | 0.3451 | 0.0224 | 0.9035 | 0.1821 |
| Mean Clusters (Traj. Transformers) | 0.3413 | 0.0325 | 0.039 | 0.0723 |
| Mean Clusters (BERT) | 0.3427 | 0.04074 | 0.8645 | 0.1098 |

Table 10: **Quantitative comparison of LSTM, Bert and Trajectory Transformers**

### C.3    Are distant trajectories really important?

**Distance State-Trajectory and its importance** Let us formally define the following variables:

- $S$: set containing all states with at least one attributed trajectory

- $T_s$: set of all the attributed trajectories for the state $s \in S$

- $t_{i,s}$: i-th trajectory in the attribution set $T_s$. Each trajectory $i$ has length $l_i$

- $a^*(b; c)$: distance from point b to point c in our grid. It is calculated by implementing the $A^*$ search algorithm.

- $d(t_{i,s}; s)$: distance from state $s$ to trajectory $t_{i,s}$. Mathematically, for each point $p_j \in t_{i,s}$,

$$d(t_{i,s}; s) = \begin{cases} 0 & : \text{ if } \exists\, p_j \in t_{i,s} \text{ s.t. } p_j = s \\ \frac{1}{l_i} \sum_{j=1}^{l_i} a^*(p_j, s) : & \text{otherwise} \end{cases}$$

- $d(T_s, s)$: Average distance of the attribution set $T_s$ from its respective state s. We implement it as:

$$d(T_s, s) = \frac{1}{|T_s|} \sum_{k=1}^{|T_s|} d(t_{k,s}; s)$$

Given the elements introduced above, we can calculate the average distance of a state from its attributed trajectories, denoted by $d(T_s, s)$. Note that in the formulation above, if a trajectory $t_{i,s}$ passes through the state $s$, we set $d(t_{k,s}; s) = 0$. This is a design choice, which can be further justified. In fact, we are interested in considering 'far' only the trajectories where there is no interaction with the state $s$ itself.

We designed and implemented Algorithm 1. We provide a high-level pseudo-code for a better understanding of the steps we perform.

---

**Algorithm 1:** Algorithm for calculating the average distance State - Attributed Trajectories

---

**1 Inputs:** $S$, $T_s$ for $s \in S$
**2 foreach** $s \in S$ **do**
**3**      D = empty list
**4**      **foreach** $t_{i,s} \in T_s$ **do**
**5**          M = empty list
**6**          **foreach** $p_j \in t_{i,s}$ **do**
**7**              point distance = $a^*(p_j, s)$
**8**              append point distance to M
**9**          **end**
**10**      **end**
**11**      m = Average of the list M
**12**      Append m to D
**13**      Set m = 0
**14 end**
**15 return the list D**. It contains the average distances of each state $s$ from its attributed trajectories.

---

We introduced a clear metric, together with an algorithm that provides details on how to compute it.

### C.4 Are data trajectories important to obtain a good action value? Are some more important than others?

In this section, we aim to provide further details on experiments on the assumptions of Claim *Removing Trajectories induces a lower Initial State Value*. Values from both the original paper and from Table 1 suggest that data trajectories are important to obtain a good ISV for our state. We are interested to see if some of the clusters are more important than others in determining this value. The setting of the experiments is equivalent to the one introduced in Section 4.3

Figure 2 shows that data trajectories play a factor in obtaining a satisfying action value. For simplicity, the original policy is not plotted, but its value is higher than any other policy in both cases. This again connects

with and proves claim *Removing Trajectories induces a lower Initial State Value*, even with a higher grid size and number of trajectories. The plot illustrates the action value for each of the policies $\pi_1, \ldots, \pi_m$ illustrated in Figure 1. For each cluster $C_i$, we show the action value obtained with the policy $\pi_i$, plotted against the number of times this cluster $C_i$ has been the responsible cluster for a change in the decision of the agent (on the x-axis). In Figure 2 we observe a clear inverse correlation between the two. Clusters that have been attributed more often to a change in decision are important in obtaining a high ISV. In fact, keeping this data out of our policy training induces a lower action value for our state. This shows that the higher the importance of the cluster, the higher the gap in performance. We believe this is an interesting result, which further indicates the conceptual importance of the trajectory attribution method. In fact, it is not only a matter of the number of trajectories we train on. We found that specific clusters can hold a larger weight in the decision of an agent. This suggests that some trajectories are more fundamental than others.

### C.5 Additional Hyper-Parameters Experiments

In this section we investigate the change in metrics when hyperparameters regarding the agent training are changed. We play with the values of *alpha*, *gamma* and *number of evaluation epochs*. We proceed to generate offline data for each combination of the above mentioned hyperparameter and successively train the authors Seq2Seq model.

The best results in terms of loss value are shown in Table 11. However while we reach better loss results we

Table 11: **Experimental Results**

| Alpha | Gamma | Eval. Epochs | Loss Value |
|-------|-------|--------------|------------|
| 0.1   | 0.95  | 15           | 0.0678     |
| 0.1   | 0.5   | 5            | 0.0965     |
| 0.1   | 0.5   | 10           | 0.0965     |
| 0.01  | 0.01  | 5            | 0.0369     |
| 0.001 | 0.1   | 15           | 0.0815     |

do not necessarily obtain better overall metrics.

## D   Training setting for *Seaquest* and *HalfCheetah*

As mentioned throughout the paper, we implemented from scratch the code for the *Seaquest* and *HalfCheetah* environments. Due to the lack of details provided in the original study, we provide our own setup for the training of the aforementioned environments.

For *Seaquest*, we train a Discrete SAC model based on the original work by Christodoulou (2019), developed by Seno & Imai (2022). Observations for the game were in the form of 84x84 greyscale frames, which we stacked, forming for each observation a 4x84x84 array. This allowed the model to incorporate some degree of temporal awareness, also referred to as *context* in the original work. Subsequently, in order to preserve spatial information, we implemented a custom encoder for the model, in the form of a Convolutional Neural Network. We are not aware whether the authors pursued this approach in their study, but due to the nature of the data itself, we are sure this implementation helped the training and performance by a significant margin. Once more, due to the nature of the data (images), we implemented a 'pixel' scaler for preprocessing purposes to act as a pixel value normalizer.

For *HalfCheetah*, on the other hand, we implemented a SAC model based on the original work by Haarnoja et al. (2018). We kept the standard hyperparameters provided in the documentation of the library by Seno & Imai (2022). One critical missing piece of information we extensively discussed upon implementation was given by the notion of difference in action between models: due to the continuous nature of the actions in this environment, it becomes almost impossible for two policies to predict the same set of actions. For this purpose, we decided to implement a way of comparing actions based through 'numpy.isclose(a,b)', by Harris

et al. (2020). The similarity formula given by the above method is

$$|a - b| \leq (\text{abs. tolerance} + \text{rel. tolerance} * |b|)$$

Unlike the built-in 'math.isclose', the above equation is not symmetric in $a$ and $b$: it assumes $b$ is the reference value – so that 'isclose(a, b)' might be different from 'isclose(b, a)'. Furthermore, the default value of 'abs. tolerance' is not zero, and is used to determine what small values should be considered close to zero. Once again, we are not aware of what the authors did on this end, but we have reasons to believe this approach has grounds for a correct interpretation. We acknowledge potential sensitivity to the results based on the choice of the hyperparameters of the method, for which we kept the default ones.

Finally, although the authors mention a training schedule *until saturation* without further explanation, we followed the guidelines provided in the DR3RLpy framework as outlined by Fu et al. (2021), training both our models for 10 epochs, each taking 100 steps. We are not aware of whether our approach matches the one suggested by the authors, but results, although not in absolute value, are relatively consistent with those provided in the original study. Thus we can conclude that our hyperparameters setting is sufficient for reproducing the results.

