# OpenReview forum: "'Explaining RL Decisions with Trajectories’: A Reproducibility Study"
_TMLR — Accepted by TMLR_

### Review · Reviewer_7zst · 2024-03-14

**Summary Of Contributions:**

This paper tries to reproduce the result from 'Explaining RL Decisions with Trajectories'.

The original paper's method consist in encoding trajectories from a training set and then to applying clustering algorithm to extract clusters of explored behaviors, and finally explains a decision made by the trained RL agent with the cluster that, by comparing with an agent trained without its trajectories, has the largest impact on that decision. Then, 3 studies are conducted to analyze these clusters:

1) A quantitative study, that measures the influence of each cluster by removing its trajectories from the training set, with many divergence metrics. ("RTISV" claim).

2) A qualitative study that displays representative trajectories of the different clusters and show how they influence a specific decision at a specific state. ("CHLB" and "DTDA" claims)

3) A human study, where they ask subjects to guess the representative trajectory that best explains a decision of the trained RL agent, to see if this corresponds to the cluster that the algorithm attributed to the decision. ("HS" claim)

1) and 2) were conducted on 3 envs: a gridworld, Seaquest, Halfcheetah, while 3) is only conducted on a gridworld.

The general observations of the reproducibility study are:
- results of 1) 2) and 3) on the gridworld are approximately reproduced (up to a seed), and claims are validated.
- results of 1) and 2) on Seaquest and Halfcheetah diverge from the original paper (which is expected since the env are slightly different and the algorithms differ a bit, but the claims can still be validated.

**Audience:**

Yes

**Broader Impact Concerns:**

No ethical concern.

**Claims And Evidence:**

Yes

**Requested Changes:**

The most important changes would be to add more environments, to repeat the quantitative experiment with multiple seeds (at least 5), to report both means and standard deviations.

Regarding the redaction, I would strongly suggest to split the paper in 4 section, one for each claim, in order to avoid the “RTISV”, “CHLB”, “DTDA” notation.

Also, the fact that the seed used in the original paper was not communicated is not relevant and is not an issue. What we expect from reproducibility is not to recover the exact values but to validate the claims.

Details:
The carbon footprint section is irrelevant and should be moved to the appendix.

**Strengths And Weaknesses:**

*Strengths*

The paper clearly explains in detail all the hyperparameters and/or provides the github links to the used code sources.

The authors conducted additive studies regarding the impact of the choice of the clustering method.

*Weaknesses*

The experiments of the original paper were particularly poor: only 3 environments including a very small griworld, only one seed for the quantitative study, only 10 subjects in the human study. Hence, we would expect a reproducibility study to explore larger scales and counterbalance this issue. But the only “correction” was to add 10 more subjects to the human study. Consequently, this reproducibility study inherits the limitations of the initial paper.

Another problem of this paper is the redaction. I found it hard to read, with:


A lot of concepts used before being explained
(for ex: what does 'removing trajectories' means in the abstract, or what does 'high level behaviour' means in section 2),

Redundant structures that repeats several time the same sentences
(for ex: the story of the seed).


The “RTISV”, “CHLB”, “DTDA” and “HS” notation, that forces the reader to go back to p.2 in order to remember what claims are addressed every time these letters are appearing.

---

> ### Author Response · Authors · 2024-04-26
> **New environments and paper redaction**
>
> Dear Reviewer 7zst,
> we thank you for your review and comments. We incorporated your suggestions to improve the paper. Here below we list the changes we made:
>
> - Implemented the Atari Breakout environment. You can see a recap of the metrics for this environment in Table 8. We added a visualization of the clusters for this environment in Figure 12.
>
> - Implemented the Q*Bert Breakout environment. You can see a recap of the results for this environment in Table 7. IWe added a
>   visualization of the clusters for this environment in Figure 12.
>
> - We repeated our quantitative metrics over 6 different seeds. We updated the new results for the metrics in Table 1 and 2. We indicated the standard deviation as well
>
> -  We removed words that referred to concepts that were not introduced yet, such as 'removing trajectories' in the abstract, or 'high level behavior' in Section 2.
>
> - The references to the seed of the original experiments have been removed.
>
> - We moved the carbon footprint section to the appendix.
>
> - The redaction of the paper has been heavily edited. As you suggested, each claim has its own dedicated section, where we report the reproducibility and additional experiments. The claims are not referred to with acronyms anymore, enhancing clarity for the reader.
>
> We thank you again for your comments

---

> ### Comment · Reviewer_7zst · 2024-05-10
> **Thanks for the improvements**
>
> The new version of the paper addresses my most important concerns, and this time brings significative gains to interpret the extent of the initial paper.
>
> However, I am still on the fence regarding the impact and the interest of this work.
>
> Also, the number of subject for the human study is still too low to be interpretable, in both the original and in that reproducibility paper.

---

> ### Author Response · Authors · 2024-05-14
> **Thank you for your answer**
>
> Dear Reviewer 7zst,
>
> We thank you for the acknowledgement of the changes and further advice.
>
> With regards to the paper impact and interest, we believe that our contributions are mainly twofold:
>
> - We implement their codebase from scratch. This will allow future people in the field to be able to conduct research using the methodology. We deemed this one as the biggest drawback from the original paper. In fact, while the method is valuable, not having a ready to use implementation ‘causes’ a bottleneck for expanding with further experimentation. Also note that very little indications from the code were given by the original authors, which caused more effort to be spent into this.
> - We expand and clarify the scope of the original paper through a thorough verification of the claims, a reproducibility study of the results and additional experiments and method to validate their findings.
>
> Additionally, we are mindful of the original paper's lack of human interpretability. Although we doubled the number of interviewees and the number of trajectories used, we agree that a more comprehensive human study could have been carried out. Given the limitations of the original paper, we were at a crossroad between the implementation of the repository or further extending on the aforementioned. We believed the implementation would have provided a bigger contribution to the research community. Nonetheless, we are aware of the tradeoff this posed and thank you for asking a clarification.
> We hope that the main contributions are considered as impactful, and that the requested changes added enough value to the paper.
>
> We thank you again for your comments.

---

### Review · Reviewer_NCRA · 2024-03-20

**Summary Of Contributions:**

The paper attempts to reproduces 'Explaining RL Decisions with Trajectories'. They partially confirm findings and extend analysis with different metrics and analysis.

**Audience:**

Yes

**Claims And Evidence:**

No

**Requested Changes:**

Perhaps the easiest way to make the paper more attractive significantly increase the empirical content. This could include:
- more enviroments
- more RL training algorithms
- study of design choices (akin to the on in Sec 4.2.1 but in a much broader scope)

etc.

**Strengths And Weaknesses:**

Strengths:

Understanding the influence of the data for agents behaviour is the key aspect of RL.

Weaknesses:
The relevance of the paper is limited. While the reproduction of result is valuable it is of limited scope. I would much love to see more environments. It is very interesting if the conclusion hold for other training methods etc. The authors barely scratch the surface of interesting questions.

---

> ### Author Response · Authors · 2024-04-26
> **New Environments**
>
> Dear Reviewer NCRA,
> we thank you for your review. Here below we report the changes we made to integrate your suggestions:
>
> - Implemented the Atari Breakout environment. You can see a recap of the metrics for this environment in Table 8. We added a visualization of the clusters for this environment in Figure 12.
>
> - Implemented the Q*Bert Breakout environment. You can see a recap of the results for this environment in Table 7. IWe added a
>   visualization of the clusters for this environment in Figure 12.
>
> - Added as a new design choice the Trajectory Transformer [1] encoder to get a different representation of the trajectories.
>
> - Added as a new design choice the BERT Transformer [2] encoder to get a different representation of the trajectories.
>
> We would like to emphasize that our room for new implementations was limited by the lack of open-source pre-trained encoders. As you can imagine, these encodings require significant amounts of compute, which was a constraint in our study. Therefore, we added two environments we found pre-trained models for, namely Breakout and QBert.
>
> We thank you again for your review.
>
>
> [1] Janner, M., Li, Q., & Levine, S. (2021). Offline reinforcement learning as one big sequence modeling problem. Advances in neural information processing systems, 34, 1273-1286.
>
> [2] Devlin, J., Chang, M. W., Lee, K., & Toutanova, K. (2018). Bert: Pre-training of deep bidirectional transformers for language understanding. arXiv preprint arXiv:1810.04805.

---

### Review · Reviewer_HzA9 · 2024-04-16

**Summary Of Contributions:**

This work presents a reproduction study of the paper "Explaining RL Decisions with Trajectories" (Deshmukh et al, 2023). The original study seeks to attribute the policy decisions of train offline RL policies to specific data subsets. These subsets are computed by clustering the trajectory embeddings—generated by pretrained encoders—and training explanation policies on each cluster, defined as the policy trained on the complementary dataset w.r.t. to each cluster. Given an offline RL policy trained on the full dataset, its action choice at a given state is then attributed to a specific cluster by finding the cluster corresponding to the explanation policy whose own action choice is furthest from the offline RL policy's action choice, in terms of some action-space metric. This original study and this reproduction study both investigate this attribution approach in three environments: Gridworld, Seaquest, and HalfCheetah.

In particular, these studies seek to find support for four phenomena:
- Removing clusters of trajectories from the training set leads to lower Initial State Value.
- Trajectory embedding clusters correspond to meaningfully different behaviors.
- Distant trajectories influence the agent decisions
- Humans can correctly identify the attributed clusters among a set of trajectories including random ones

The study finds many of these results are reproducible, though not consistently so across all environments.

**Audience:**

Yes

**Broader Impact Concerns:**

The conclusions of this paper somewhat strengthen the claims in the original Deshmukh et al, 2023 paper, which introduces a method for attributing the decisions made by an offline RL policy on specific clusters of trajectories. This work could lead to such attribution methodologies be adopted more broadly. This poses a risk if such attribution methods result in misleading or spurious attributions in real-world settings. This is a general concern for many interpretability methods.

**Claims And Evidence:**

Yes

**Requested Changes:**

Following the points raised in the weaknesses section, I would suggest the following changes:
- Add more motivation in the intro for why the authors believe it is particularly important to reproduce Deshmukh et al, 2023.
- Add a deeper discussion on the key takewaways from this reproducibility study, as well as on the limitations of the attribution method and the settings studied.
- Include a skimmable summary (e.g. a table) of the key results: Which original results were reproduced and which were not, as well as methodological differences. The latter might be a separate listing in the appendix.
- A brief description for why DBSCAN is a sensible alternative method to try in addition to the original X-Means method for clustering would improve the legibility of the experiment design.
- The carbon footprint calculations break the flow of the paper significantly. I suggest moving this to the appendix.
- The abbreviations for the main phenomena under study, e.g. RTISV, are not memorable and make the paper harder to read. I suggest renaming these to short terms more naturally indicative of each phenomenon.

**Strengths And Weaknesses:**

### Strengths
- This study places a lot of effort in comprehensively reproducing the original Deshmukh et al, 2023, including running a human study. The authors of this submission clearly placed a lot of attention to faithfully reproduce every detail of the prior work.

- The authors seek to improve the method of analysis for some of the phenomena of study, such as investigating the use of an alternative clustering algorithm, DBSCAN in place of X-Means, as well as seeking to understand how varying the amount of trajectories used in clustering impact the resulting clusters. Moreover, the authors also provide a more quantitative measure on which to base the finding on the idea that an agent's behavior can be attributed to distant states, by computing an average distance of each attributed state to the trajectory states in its attributed cluster.

- The authors convincingly reproduce several of the empirical results attained by Deshmukh et al, 2023.

### Weaknesses
- It is not clear what can be learned from the observation that training on all trajectories leads to an improved policy. Can this be due to the higher-return trajectories being more useful for learning? Ablations on the specific data mixture would be more informative here.

- While the authors make a commendable effort in describing exactly the differences in their reproducibility study from that of the original, it would benefit the paper to include a skimmable summary of these differences, as well as which empirical results were reproduced and which were not.

- The biggest weakness is that the authors are not clear on the main takeaways from their work. Given the choice to reproduce any offline RL paper, why choose this one?

- Related to the above, the authors should elaborate on the conclusions that can be drawn from the results of their reproducibility experiment. What are the limitations of this approach for trajectory attribution in general, e.g. when might there be false positives or false negatives in attribution, how might the constraint on single-cluster attribution limit the nature of the resulting attributions? What are the possibilities opened up by such interpretability methods, should they be consistently reproducible?

---

> ### Author Response · Authors · 2024-04-26
> **Paper Redaction and new details**
>
> Dear Reviewer  HzA9,
> thank you for your detailed review and feedback. We first address your questions:
>
> Question 1: It is not clear what can be learned from the observation that training on all trajectories leads to an improved policy. Can this be due to the higher-return trajectories being more useful for learning? Ablations on the specific data mixture would be more informative here.
>
> Answer 1: Here we aimed to validate the claim that trajectories are indeed important for the policy to converge. If we train without some determined clusters of trajectories, and indeed our state value decreases, it means that these were not redundant and indeed useful to obtain a satisfying policy. We are not sure what you refer to with 'data mixture' itself. Could this be clarified?
>
> Question 2: Given the choice to reproduce any offline RL paper, why choose this one?
>
> Answer 2: We deemed particularly valuable the novel contribution from the authors to propose a new framework for explainibility in RL. At the same time, we noticed the lack of an available implementation from their work. Hence, we thought it could be of scientific interest to implement their environments from scratch, hoping to make this framework more accessible to other researchers.
>
> Question 3: What are the limitations of this approach for trajectory attribution in general, e.g. when might there be false positives or false negatives in attribution, how might the constraint on single-cluster attribution limit the nature of the resulting attributions? What are the possibilities opened up by such interpretability methods, should they be consistently reproducible?
>
> Answer 3: We added some extra information about the key takeaways of the paper as well as what we believe the attribution process could lead up to in future research.
> We did think that restraining on single-cluster attribution limits the nature of the resulting attributions, and we included it in the final analysis. We are not sure, but if your question regards the possibility that there might be attributed clusters within there are trajectories that do not necessarily suggest the chosen action, then we have an answer for you. We had already investigated this and, found that in all cases each attributed cluster is comprised of meaningful trajectories to the chosen action. However, we did not find this meaningful enough to include it in the final draft of the paper.
>
> Regarding your requested changes, here below we list the edits that incorporate them:
>
> - We added motivation in the intro explaining why we deem this reproducibility study as of value.
>
> - We added a paragraph in Section 5, "Key takeaways and original paper limitations", in which we try to summarise the limitations of our study as well as what we believe the attribution process of the paper could lead up to.
>
> - We added a skimmable summary of our results in Table 4. We strictly followed the methodology of the authors regarding the experiments. However, they provide little details in some parts, and hence we are not able to indicate in a table where our methodology exactly differs.
>
> - In Section 4.5 we added a brief discussion on why DBSCAN clustering is a relevant and sensible alternative to XMeans.
>
> - We moved the carbon footprint section to the appendix. Moreover, we restructured our manuscripts to avoid the use of acronyms as in the original submission. Now, each claim has a dedicated section where the reproducibility and additional experiments are reported.
>
> We thank you again for your review and for your comments which helped to improve the paper

---

### Decision · Action_Editor_wHma · 2024-05-17

**Recommendation:** Accept as is

**Comment:**

NCRA: "Solid empirical work with applicational relevance."
7zst: "The new version of the paper addresses my most important concerns, and this time brings significative gains to interpret the extent of the initial paper."

Reviewers 7zst has concerns about the novelty and the significance of the work. However, I believe that the paper still provides a valuable companion to the original paper, in particular since it finds insufficient evidence for one of the main claims in the original paper (iv).

Another reproducibility of the same paper appeared in TMLR: https://openreview.net/forum?id=JQoWmeNaC2 at the same time. The two studies are complementary, and given that two independent groups went on to reproduce the paper, provides an evidence of the community's interest.

**Audience:**

The paper is of interest to the RL community.

**Claims And Evidence:**

The paper reproduces, with additional empirical results, "Explaining RL Decisions with Trajectories" (Deshmukh et al, 2023). The results partially support thee main claims in the paper (fewer trajectories induce a lower initial state value, clusters of trajectories present similar high-level patterns, and distant trajectories influence agent decisions), where there is insufficient evidence for the claim that the humans correctly identify the attributed trajectories to the decision of the agent.